# Warming induces short-term phenological shifts in pollinator-plant interactions that enhance larval development in honey bee

Megan M. Y. Chang[1], Pei-Shou Hsu[2], En-Cheng Yang[3], Syuan-Jyun Sun[4]*, Chuan-Kai Ho[1]*

1 Institute of Ecology and Evolutionary Biology, National Taiwan University, Taipei City, Taiwan, 2 Miaoli District Agricultural Research and Extension Station, Ministry of Agriculture, Taipei City, Taiwan, 3 Department of Entomology, National Taiwan University, Taipei City, Taiwan, 4 International Degree Program in Climate Change and Sustainable Development, National Taiwan University, Taipei City, Taiwan

* sjs243@ntu.edu.tw (SJS); ckho@ntu.edu.tw (CKH)

**Data Availability Statement:** All data are available in the main text or the supplementary materials, and have been deposited at figshare (https://doi.org/10.6084/m9.figshare.25289134.v1).

## Abstract

Climate warming can precipitate mismatches in plant-pollinator interactions by altering their phenologies of both parties, impacting ecosystem services. While most studies have focused on long-term, seasonal phenological shifts, the effect of warming on short-term phenological match-mismatch in these interactions remains unclear. Here, we investigate how experimental warming affects within-day foraging behavior of the honey bee (*Apis mellifera*) and plant anthesis, and whether the resulting changes in bee pollen composition, in terms of the relative abundance of pollen from different plant species, influences larval development. Experimental warming advanced both the within-day foraging by bees and anthesis of *Bidens pilosa*—the predominant pollen source among all plant species represented in the collected pollen. Through experimental manipulation of pollen composition, we demonstrated that an increased proportion of *B. pilosa* pollen in the diet enhanced bee larval growth efficiency. Overall, our study demonstrates that warming may influence pollinator interactions with the many plant species by affecting pollinator behaviors and plant anthesis on short-term temporal scales, with potential implications for pollinator larval development.

## Introduction

Global environmental change induced by anthropogenic activities has major consequences for biodiversity and ecosystems [1, 2], particularly by altering species interactions [3, 4]. Among these interactions, plant–pollinator interactions are of particular significance because they provide key ecological functions, such as pollination, biodiversity support [5], and ecosystem stability [6]. Global warming can disrupt crucial plant-pollinator interactions by causing phenological mismatches [7–9]. Such disruptions could diminish the effectiveness of pollinators whose services are vital for the reproductive success of over half of all plant species [10, 11]. A reduction in effective pollination could reduce seed set in plants that rely on pollinator outcrossing for reproduction, thereby reducing the total quantity of pollen and nectar rewards

**Funding:** National Science and Technology Council 108-2621-B-002-003-MY3 (CKH) National Science and Technology Council 111-2621-B-002-003-MY3 (CKH).

**Competing interests:** All authors declare that they have no competing interests.

available to pollinators in future seasons [12]. Past studies have found that climate warming accelerates the emergence of plant flowering and pollinator foraging, especially in spring [13]. If these phenological changes occur independently between species, the timing of plant flowering and pollinator activity may no longer coincide or be reduced, disrupting the crucial interactions required for effective pollination. For example, warming can disproportionally drive a seasonal shift in activity time by inducing an earlier flowering *Corydalis ambigua* compared to the initial activity of its pollinator bumble bees. This mismatch may reduce the amount of pollination and ultimately lead to a decline in seed set [14].

Bees (Hymenoptera: Anthophila) are one of the most ecologically and economically important pollinators worldwide [15–17]. Particularly, honey bee (*Apis mellifera*) appears to be the most important, single species of pollinator across the natural systems studied, owing to its wide distribution, generalist foraging behavior and competence as a pollinator [15]. To date, most studies on phenological match-mismatch mainly focus on changes observed over the course of entire seasons, examining long-term appearance patterns of both plants and pollinators [18]. However, the impact of warming on plant-pollinator interactions within shorter timeframes, such as within-day phenological match-mismatch, remains understudied. Past research on *A. mellifera* points out that foraging behavior can change immediately in response to high temperatures [19, 20], such as decreasing foraging activity [21] or causing bees to seek water for use in evaporation cooling [22]. The foraging activity of *A. mellifera* typically begins in the early morning and finishes in the evening, and can fluctuate according to within-day variations in environmental factors, including temperatures [23]. In a closely related species, *Apis cerana*, the number of workers collecting pollen and nectar decreased with increasing temperature at a temperature range between 27˚C and 34˚C [24]. However, at a temperature range between 14˚C to 21˚C, a higher ambient temperature had a positive effect on the number of bees leaving the hive [24]. While the short-term effect of temperature change on pollinator foraging behaviors has drawn some attention, its implications for pollinator-plant interactions, such as pollinator growth or plant seed set rate, remain understudied. These implications may depend on changes in the timing of pollinator visits and/or anthesis of plants within a day [25].

In addition, to predict pollinators' responses to temperature change, it is crucial to understand how these changes affect pollen composition, i.e., the variety of bee pollen collected by bees from different plant species. This is particularly relevant because shifts in temperature may alter plant phenology, subsequently changing the timing and nature of pollen available to foragers. Such phenological shifts may influence the foraging behaviors of honey bees, who choose patches of flowers based on plant anthesis in their environment [26], ultimately affecting the diversity of their pollen loads. Moreover, circadian organization in both bees and flowers—timing of visits, petal opening, scent, and nectar availability—play a crucial role in synchronizing plant-pollinator interactions [27]. Temperature-driven shifts may disrupt these rhythms, impacting pollination networks and potentially altering community structure [27]. Such disruptions can affect not only the timing of pollen availability but also the diversity of pollen collect, which is critical for the nutritional balance of the bees' diet. A diversified pollen diet is known to provide a more balanced nutrient profile [28], which is critical not only for Click or tap here to enter text.larval development [29] but also for determining adult bee physiology [30] based on their early pollen intake. However, a study that experimentally manipulated the composition of pollen diet in bumble bees suggests that the nutrient contents of pollen of a particular species may be more important for colony development than the diversity of the pollen per se [31]. Understanding the interplay between climate-driven phenological changes and pollen composition is therefore essential for assessing the broader impacts on pollinator health and ecosystem stability.

Here, we investigate how elevated temperatures affect plant-pollinator interactions on a daily basis and the ecological consequences for the honey bee *A. mellifera*. We aim to address three main questions with three experiments: 1) does warming affect bee foraging behaviors (Experiment 1), 2) does warming affect plant anthesis (Experiment 2), and 3) how do warming-mediated changes in pollen composition influence larval development of bees (Experiment 3). We hypothesized that 1) warming would affect bee foraging behavior by altering ambient temperature within the day. That is, warming would increase the number of foragers and the weight of pollen loads carried by bees when the ambient temperature is less than optimum range, and decreases them when the ambient temperature is greater than optimum range (sensu Reddy et al., 2015); that 2) warming would affect the time of anthesis within the day, making plants begin anthesis earlier; and that 3) warming would affect bee larval development by altering the composition of pollen based on changes in bee foraging behavior and plant anthesis. Specifically, we expected to find a positive correlation between the diversity of bee pollen and the weight of bee larvae.

## Materials and methods

### Experimental design

The field experiment was conducted at National Taiwan University Farm, Taipei, Taiwan, from September 16 to November 19, 2020. Two beehives purchased from local beekeepers were used, and were kept apart by 15 meters to minimize interference between the colonies. To maintain the colonies, we fed ad libitum sugar water and pollen substitutes (a mixture of bee pollen, soybean powder, and water) in the hive once a week. To simulate warming, we used a heating mat (15×28cm, 5v, 2A, Rep-Shop®) to continuously heat one of the beehives for 20 hours on alternate days. Meanwhile, we used ibutton temperature loggers (DS1923L, Thermochron iButton) to record the temperature change inside the beehives every 5 minutes throughout the experiments. The warming treatment resulted in an average of 2.4˚C increase of beehive temperatures compared to the control group ($\chi^2$ = 146.93, df = 1, $p < 0.001$; Fig 1), resembling the future global warming scenario predicted by the end of the century (IPCC). In addition, ambient temperature was recorded by the meteorological instrument next to the National Taiwan University Atmospheric Sciences (Taipei, Taiwan). The experiments were not conducted during rainy periods because bees remained largely inactive.

**Experiment 1: Effect of warming on bee foraging behavior.** *Change of foraging behaviors.* To quantify foraging behaviors, we recorded the behaviors with a camera (Canon IXUS 285 HS (20.2 mega pixels)) positioned from the top of the landing platform (S7 Fig in S1 File). We calculated the number of foragers departing from the landing platform after exiting the beehive entrance for each beehive. The observations were sampled from 5-minute videos recorded every one to two hours from 6:00 to 16:00, aligning with the peak activity period of the bees. This schedule resulted in eight 5-minute observations each day of the experiment (a total of 26 days). To ensure consistency, these videos were recorded simultaneously for both bee hives at each time point. If bees were observed conducting orientation flights (i.e., locating beehives), that specific 5-minute observation was excluded from the analysis as these behaviors do not directly contribute to foraging activities. To determine population size, we opened the beehives every week and recorded the number of bees on each side of beehive foundations (four in total per beehive) by taking photos and analyzing them with ImageJ software (https://imagej.net/ij/download.html; [32]).

In addition to the number of foragers, we also recorded the weight of bee pollen loads as an indicator of bee foraging behaviors. Two pollen traps, designed to allow bees to pass through while scraping off the bee pollen from their legs, were used to collect bee pollen loads from the

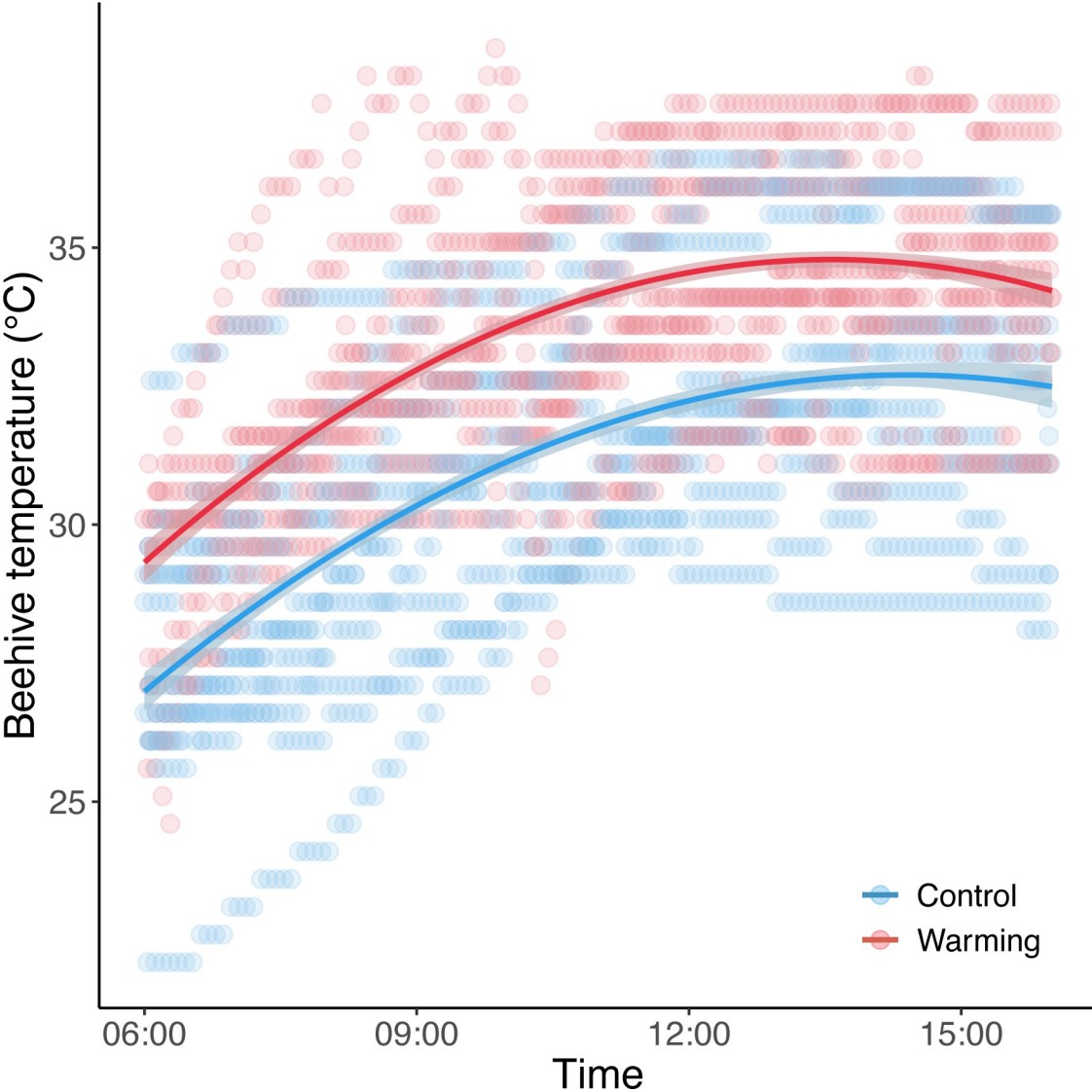

**Fig 1. The effect of warming treatment on beehive temperature over the time of day.** Lines are predicted relationships from generalized linear mixed models.

foragers of each hive. Pollen loads were collected hourly from 6:00 to 10:00, and once every two hours from 10:00–16:00, resulting in seven pollen load samples per day of the experiment. Bee pollen loads of each observation period were weighed to the nearest of 0.1 mg, and the values were adjusted to an hourly standard for further analyses. To determine the pollen collection rate per individual, we divided pollen weight by the number of returning foragers recorded from each 5-minute footage. This metric reflects the overall pollen return but does not account for non-pollen foragers (e.g., those collecting nectar or water), as only returning foragers were visible in the footage without differentiation of resource type. Once weighed, the bee pollen was stored in a freezer at -20˚C for species identification and Experiment 3 (see below).

*Pollen loads composition and species identification.* To identify plant species from which bees had collected pollen, we sampled the previously stored pollen of the same season from

one beehive and classified them based on their external morphology using naked eye observation with the aid of a compound microscope (ZEISS AX10 Scope.A1, Germany). This allowed us to make preliminary species classification according to morphological and color traits of pollen grains.

We determined the weight of each morphologically distinct pollen grain using an analytical balance (AB104-S, Mettler Toledo) to the nearest 0.1 mg. Pollen of the same species that reached more than 5% of the total pollen weight in a given sampling period of the day were identified. The top ten plant species (comprising 83.4% of the total pollen weight; S1 Table in S1 File) were then further identified using molecular analyses. To validate these species with molecular investigation, we extracted 100 mg of each of the top ten pollen samples using the Plant Genomic DNA Purification Kit (Protech, Taiwan), following the instruction manual. The sequences were amplified by PCR using the extracted DNA as a template and employing the universal primer pairs of rbcL and trnH-psbA. The primer pairs used for rbcL were F: 5′‑ATGTCACCACAAACAGAGACTAAAGC‑3′ [33] and R: 5′‑ATGAATGTCTACGCGGTG GACT‑3′ [34]; the primer pairs used for trnH-psbA were F: 5′‑GTTATGCATGAAC GTAATGCTC‑3′ [35] and R: 5′‑ CGCGCATGGTGGATTCACAATCC‑3′ [36]. The PCR reaction settings for rbcL and trnH-psbA were identical, but were processed and analyzed separately. The total volume of each PCR reaction was 50 μl, including 1 μl of F primer, 1 μl of R primer, 10 μl of Fast-RunTM Taq 5x Master Mix (Protech, Taiwan), 4 μl of pollen sample DNA and 34 μl of sterilized water. Reaction conditions were (1) 9 minutes at 94˚C, (2) 40 cycles of 30 seconds at 90˚C, 30 seconds at 58˚C and 40 seconds at 72˚C, and (3) 7 minutes at 72˚C. After the reaction, the PCR product was electrophoresed on 1.5% agarose gel, and the presence of the product and the molecular weight of the product were observed under ultraviolet light, and then the product sample was sent to a commercial company (Genomics, Taiwan) for Sanger sequencing analysis. The bidirectional sequence was then manually aligned and debugged with Clustal Omega [37] software. The primer fragments were removed, yielding the corrected rbcL and trnH-psbA sample sequences, and BLASTn [38] software was used to compare the similarity with the GenBank database sequences to identify plant species.

**Experiment 2: Effect of warming on the time of anthesis.**   In this experiment, *Bidens pilosa* (Asteraceae) was selected as the object to study the time of plant anthesis because it had the highest abundance of all bee pollen (comprising 28.0% of the total bee pollen weight; S1 Table in S1 File), and could be harvested throughout the period of the field experiment (S3A Fig in S1 File). The seeds of *B. pilosa* were randomly collected from the campus of National Taiwan University and planted in a growth chamber with a constant temperature of 28˚C and a constant humidity (60% RH) under a 12/12-hour day/night cycle.

On the 17th day of seedling growth, when most individual plants had grown the second trifoliate leaf, they were moved to temperature-controlled greenhouses in the phytotron with natural light. We randomly assigned individuals to two temperature-controlled greenhouses for continued cultivation according to the ambient day-time temperature (26.7˚C; 6:00–19:00) and night-time temperature (24.0˚C; 19:00–6:00) in the field experiment. The day and night temperature of the control group was 25/20˚C, while the day and night temperature of the warming group was 30/25˚C. During the cultivation period, the individual positions were adjusted appropriately so that each individual was exposed to natural sunlight. The soil was kept moist, with growth fertilizer supplied once a week; Huabao No. 2 and No. 3; HYPO-NeX® was used to promote plant growth and bud maturation, respectively. After another 38 days, the observation started when individuals in both the control and the warming group had the head with mature disk florets. Individuals that had just started their first bloom or finished their last bloom were not used. The experiment included a total of 39 individuals in the control group and 30 individuals in the warming group (In total, n = 832 and 834 disk florets were

sampled in the control and warming group, respectively). From 6:30 am to 12:00 pm, the number of disk florets with style elongated before style branches during anthesis (hereafter "number of florets at anthesis"; see also [39]) was recorded every half an hour. Style elongation was used as a criterion because it marked the point at which *B. pilosa* pollen became available for collection by bees. After each observation, the disk florets alongside their stamens were removed with a pair of tweezers to facilitate the next observation.

**Experiment 3: Consequence of pollen composition change on bee larval development.** To investigate warming-mediated effects on pollen composition and its fitness consequences on bee larval development, we experimentally manipulated the composition of bee pollen loads and fed them to bee larvae. The pollen loads were collected between 11–19 November 2020, during which the field experiments were conducted. Four different pollen composition treatments were formulated, combined with the semi-artificial diet modified from Hendriksma et al. (2011) [40], and fed to the larvae: (1) the control group (*B. pilosa*: others = 38%: 62%), (2) the warming group (*B. pilosa*: others = 47%: 53%), (3) the warming plus group (*B. pilosa*: others = 55%: 45%) and (4) the *B. pilosa* only group (*B. pilosa*: others = 100%: 0%) (see Fig 4A and S2 Table in S1 File for complete percentage). These pollen compositions were determined according to the formula:

$$P_c = \frac{\sum_{t=1}^{i} W_t B_t}{\sum_{t=1}^{i} W_t}, \qquad P_w = \frac{\sum_{t=1}^{i} W_t B_{t+1}}{\sum_{t=1}^{i} W_t}$$

$P_c$ was the weight percentage of *B. pilosa* pollen in the control group, whereas $P_w$ was the weight percentage of *B. pilosa* pollen in the warming group. $t$ was a certain time of day from 6:00 to 16:00 (a total of 7 time periods). $W_t$ was the average weight of bee pollen given a certain $t$. $W_t$ for both $P_c$ and $P_w$ were identical for all time periods except for that of 6:00, according to our pollen weight results. $B_t$ was the weight percentage of *B. pilosa* in a certain time of day. The weight percentage of *B. pilosa* pollen in the warming plus group was equaled to $P_w+(P_w-P_c)$. While the percentage of other plants varied between groups, we adjusted the pollen weight of each species according to their ratio from the control temperature treatment.

A side of hive foundation with sufficient eggs was chosen. We fixed the transparent plastic sheet on top of the hive frame and used a marker to record the position of each egg on the plastic sheet. To ensure similar developmental stage of bee larvae, we selected newly-hatched larvae the next day. Two days later, the hive frame containing 3$^{rd}$ instar larvae (n = 82) was brought back to the laboratory for Experiment 3. According to previously established protocol, all larvae were maintained in climate-controlled environments and fed with the semi-artificial diet modified from Hendriksma et al. (2011) [40].

A saturated solution of potassium sulfate ($K_2SO_4$) was made and added to the storage box to keep the humidity stable, with an iButton temperature logger placed inside to monitor the temperature and humidity. The temperature was maintained at 35±0.25°C and the humidity was maintained at 95±3%. The semi-artificial diet included two different food formula weight ratios fed to the larvae as they grew. On the first day when the 3rd instar larvae were moved to the growth chamber, the larvae were fed with the diet containing 3% bee pollen, 47% royal jelly and 50% aqueous solution (3% yeast extract, 15% glucose, 15% fructose and 0.2% Nystatin (50mg/ml; SIGMA; Biobasic). Among them, Nystatin was used to prevent the pollen in the formula from being infected by fungi [41]. From the second day onwards, the larvae were fed with the diet containing 3% bee pollen, 47% royal jelly and 50% aqueous solution, with the aqueous solution formula adjusted to: 4% yeast extract, 18% glucose, 18% fructose and 0.2% Nystatin (50mg/ml). The prepared diets were divided into small tubes and stored in a -20°C freezer, and were thawed and warmed to 35°C before feeding to larvae.

Following the methodology of Hendriksma et al. (2011), each larva was fed a semi-artificial diet over four consecutive days, receiving increasing volumes of 20, 30, 40, 50 μL on each successive day, respectively. To accommodate individual variations in feeding status, we divided the daily diet volume into 4–6 portions. These portions were then fed to the larvae during the daytime at intervals of approximately three hours, and the volume consumed during each feeding was recorded. When the 5th instar larvae curled vertically to the plane of the hive foundation (prior to the prepupal stage), the larvae from the brood comb were removed using a pair of tweezers and weighed to the nearest 0.1 mg. We then determined the efficiency index of bee larval development by dividing their body mass (mg) by the actual diet volume consumed (μL).

## Statistical analysis

All statistical analyses were conducted using the statistical software R version 4.1.1 [42]. Generalized linear mixed models (GLMMs) were conducted in the package *lme4* [43] to analyze the effects of warming on bee behavior, plant anthesis, and larval development. AIC-based model selection was performed to compare models with different sets of explanatory variables, including whether to include a polynomial term. Models were evaluated based on their AIC values, and the statistical results were reported based on the best-fitting models. For all comparisons, $p$ values < 0.05 were considered statistically significant. No multicollinearity was detected in either model under model selection, with all VIFs (Variable Inflation Factors) < 5 [44]. All graphs were prepared in the package *ggplot2* [45].

**Bee foraging behavior.** To investigate whether warming affected bee performance, we analyzed in GLMMs number of foragers with Poisson error distribution, whereas bee pollen weight and pollen collection rate per individual with Gaussian error distribution. Pollen weight and pollen collection rate per individual were log-transformed prior to analyses to meet the assumption of normal distribution. In all models, we included the fixed effects temperature treatment (0: control, 1: warming) as a categorical variable, the time of day as a continuous variable, and their interaction, while date nested within bee hives was included as a random effect. To account for potential non-linear relationships predicted by the time of day, a third-degree polynomial was included initially but was reduced to lower degrees when appropriate based on model selection using AIC values. For analyses of number of foragers and bee pollen weight, we also included ambient temperature and colony size (scaled and centered) as continuous fixed effects. Once a significant interaction between warming and the time of day was detected, we proceeded to analyze the data at each time point separately. This involved fitting GLMMs for the data collected at each time interval, allowing us to assess whether the temperature treatment had a significant effect at specific times of day.

**Pollen composition.** To compare the composition of pollen collected by bees under control and warming scenarios, we used non-metric multidimensional scaling (NMDS) using the function metaMDS, with temperature treatment and the time of day as fixed effects by using the *vegan* package [46]. NMDS was based on Bray–Curtis dissimilarity to calculate the distance matrix for an ordination with 999 iterations. When applying NMDS, we constrained permutations within groups of days based on similar bee pollen composition (see S3A Fig in S1 File).

**Plant anthesis.** To investigate whether warming affects plant anthesis, we analyzed the number of florets at anthesis using a Poisson GLMM by including temperature treatment (0: control, 1: warming), the time of day, and their interaction as fixed effects. Similarly, we included the second degree of the time of day to account for potential non-linear relationships. *B. pilosa* individual identity was included as a random effect since each individual was sampled every 30 minutes for a total of 12 observations.

**Larval development.** To test for the effect of warming-mediated pollen composition on larval development, we analyzed the efficiency index (larvae weight divided by fed diet; log-transformed) in a GLMM with Gaussian distribution. Pollen composition (i.e., control group, warming group, warming plus group and B. only group) was included as a categorical variable. We then conducted Tukey post-hoc comparisons using the lsmeans package [47] to test for differences in larval development between different pollen composition.

## Results

### Experiment 1: Effect of warming on bee foraging behavior

**Foraging behaviors.** We found that the number of foragers departing from beehives was the highest in the early morning and decreased with time ($\chi^2$ = 40.30, df = 1, $p$ < 0.001), but the trend differed between the temperature treatment groups (Time$^3$ × Warming: $\chi^2$ = 3.98, df = 1, $p$ = 0.046; Fig 2A; Table 1). Further analyses for each time point suggested that at 6:00, warming induced more foragers to depart from beehives compared to the control temperature ($\chi^2$ = 4.00, df = 1, $p$ = 0.046; S1A Fig in S1 File). Similar patterns were found in the weight of pollen loads carried by bees, with the highest total weight of bee pollen loads recorded in the early morning and a subsequent decrease over time ($\chi^2$ = 33.31, df = 1, $p$ < 0.001). These patterns were mediated by the temperature treatment (Time × Warming: $\chi^2$ = 4.63, df = 1, $p$ = 0.031; Fig 2B; Table 1). Again, it was at 6:00 that the warming treatment increased the weight of pollen loads ($\chi^2$ = 5.06, df = 1, $p$ = 0.024; S1B Fig in S1 File). Furthermore, the pollen collection rate per individual was the highest in the early morning and decreased over time ($\chi^2$ = 11.16, df = 1, $p$ < 0.001; S2 Fig in S1 File; Table 1) and was not significantly different between temperature treatment groups ($\chi^2$ = 0.27, df = 1, $p$ = 0.603).

**Composition of bee pollen loads.** To assess the diversity and timing of pollen collection by bees, we analyzed the composition of bee pollen loads. From these loads, we identified 75

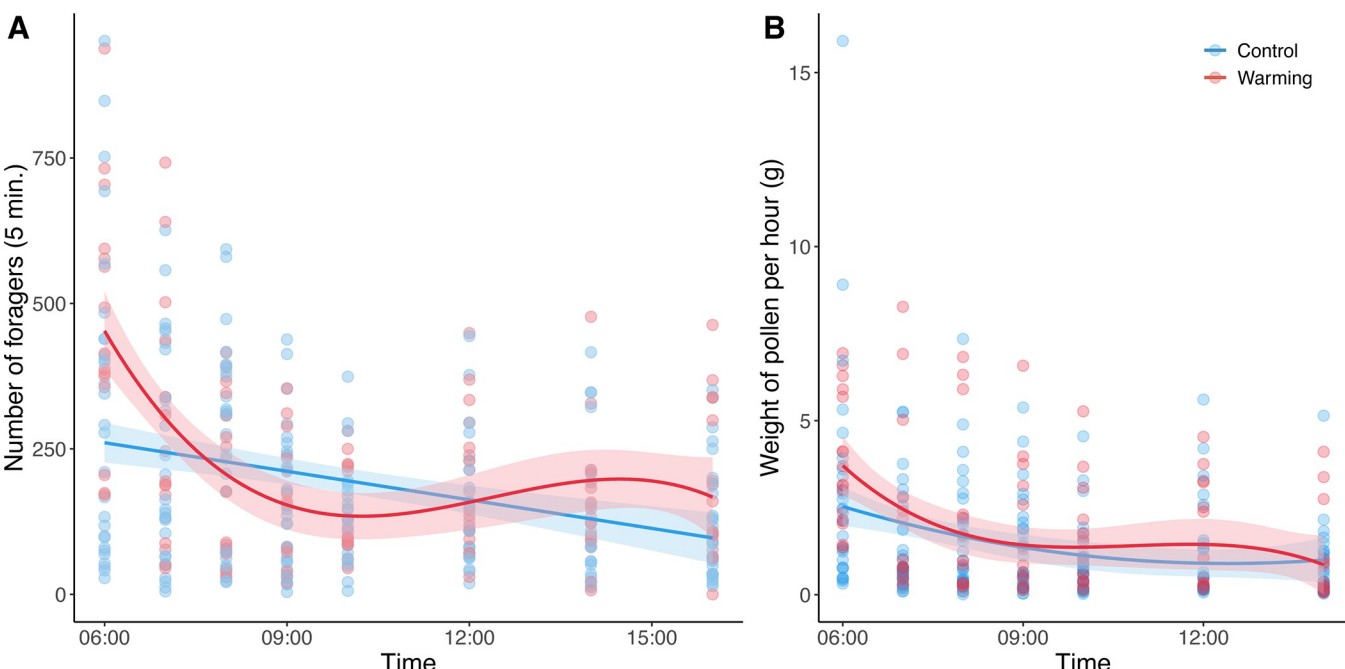

**Fig 2. The effect of temperature treatment on bee performance.** Bee performance was measured as (A) the number of foragers departing from beehives and (B) the weight of pollen loads over the time of day. Lines are predicted relationships from generalized linear mixed models.

**Table 1. Results of the ANOVAs for bee foraging, plant anthesis and larval development.**

| Dependent variable | Explanatory variables | $\chi^2$ | df | $p$ value |
|---|---|---|---|---|
| Beehive temperature | Temperature treatment | 146.93 | 1 | **<0.001** |
| Number of foragers | Time | 40.30 | 1 | **<0.001** |
| | Time$^2$ | 4.21 | 1 | **0.040** |
| | Time$^3$ | 0.44 | 1 | 0.507 |
| | Temperature treatment | 0.08 | 1 | 0.783 |
| | Ambient temperature | 0.72 | 1 | 0.397 |
| | Colony size | 93.11 | 1 | **<0.001** |
| | Time$^2$ × Temperature treatment | 4.17 | 1 | **0.041** |
| | Time$^3$ × Temperature treatment | 3.98 | 1 | **0.046** |
| Pollen weight | Time | 33.31 | 1 | **<0.001** |
| | Time$^2$ | 14.60 | 1 | **<0.001** |
| | Time$^3$ | 9.69 | 1 | **0.002** |
| | Temperature treatment | 3.69 | 1 | 0.055 |
| | Ambient temperature | 4.36 | 1 | **0.037** |
| | Colony size | 9.58 | 1 | **0.002** |
| | Time × Temperature treatment | 4.63 | 1 | **0.031** |
| Pollen collection rate | Time | 11.16 | 1 | **<0.001** |
| | Temperature treatment | 0.27 | 1 | 0.603 |
| | Ambient temperature | 0.57 | 1 | 0.448 |
| Number of flowers | Time | 54.15 | 1 | **<0.001** |
| | Time$^2$ | 87.99 | 1 | **<0.001** |
| | Temperature treatment | 28.84 | 1 | **<0.001** |
| | Time$^2$ × Temperature treatment | 14.44 | 1 | **<0.001** |
| Efficiency index | Different pollen composition | 19.46 | 3 | **<0.001** |

$p$ values <0.05 are highlighted in bold.

different plant species from bee pollen loads based on their morphological traits (S3 Fig in S1 File). The species with the ten highest relative abundance during our study period were further identified molecularly, with *Bidens pilosa* as the highest (28.0%), *Koelreuteria elegans* as the second highest (12.3%), and *Bauhinia x blakeana* as the third highest (9.8%) (see S1 Table in S1 File for other species and proportions). While pollen from different plant species were collected periodically, *B. pilosa* pollen was consistently collected by bees every day throughout our study (S3A Fig in S1 File), leading to its selection as the subject in our plant anthesis experiment (see Methods). The composition of pollen varied by time of day. Between 6:00 and 7:00, pollen of *B. pilosa* represented an average of 0.77% of pollen loads collected by bees, but after 8:00, *B. pilosa* pollen increased markedly, representing an average of 65.08% (S3B Fig in S1 File). Furthermore, NMDS results showed that the composition of bee pollen loads varied with the time of day ($F = 5.71$, $p = 0.001$) but not with the temperature treatment ($F = 1.12$, $p = 0.342$; S4 Fig in S1 File).

## Experiment 2: Effect of warming on the time of plant anthesis

The time of *B. pilosa* anthesis was affected by the temperature treatment, although the effect varied through time (Time$^2$ × Temperature treatment: $\chi^2 = 14.44$, df = 1, $p < 0.001$; Table 1). Specifically, warming advanced *B. pilosa* anthesis by one hour earlier than the control (i.e.,

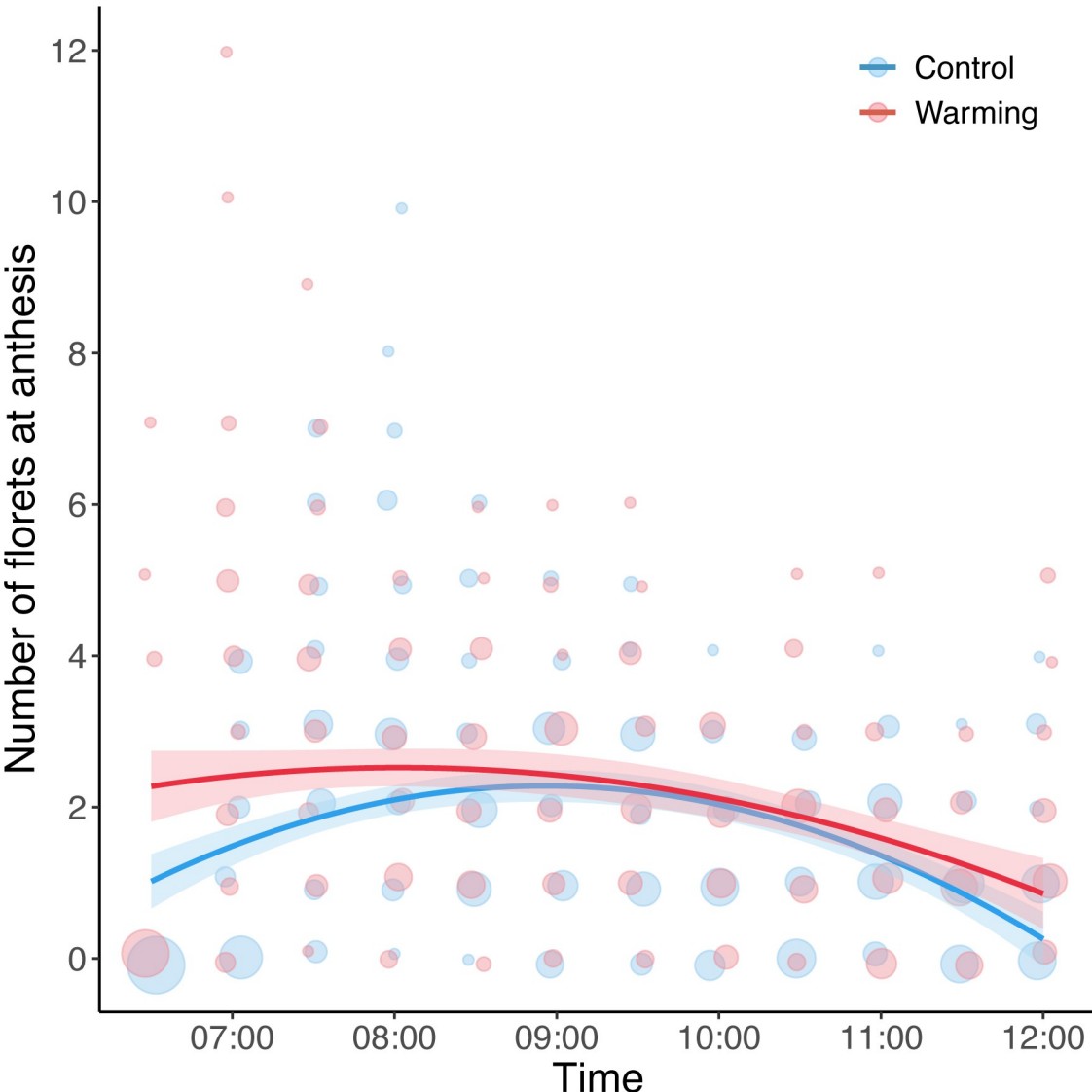

**Fig 3. The effect of temperature treatment on the number of florets at anthesis over time.** Point size represents the total number of florets at anthesis over the time of day. The peak of floret numbers was 9:00 under control treatment (25/20˚C), while the peak was 8:00 under warming treatment (30/25˚C). Lines are predicted relationships from generalized linear mixed models.

peaking at 8:00 vs. 9:00) (Fig 3). Warming overall also resulted in a higher number of florets at anthesis compared to the control (Table 1).

### Experiment 3: Consequence of pollen composition change on bee larval development

Since warming could drive earlier *B. pilosa* anthesis, thus altering the pollen that is available (Experiment 2), we experimentally manipulated the composition of bee pollen loads to be fed to bee larvae. The results showed that increased pollen proportion of *B. pilosa* significantly increased larval development efficiency ($\chi^2$ = 19.46, df = 3, $p$ < 0.001; Table 1). Post-hoc comparisons revealed that the warming, warming plus and *B. pilosa* only treatment groups all had higher efficiency indexes of larvae than the control group (Fig 4B; Table 2).

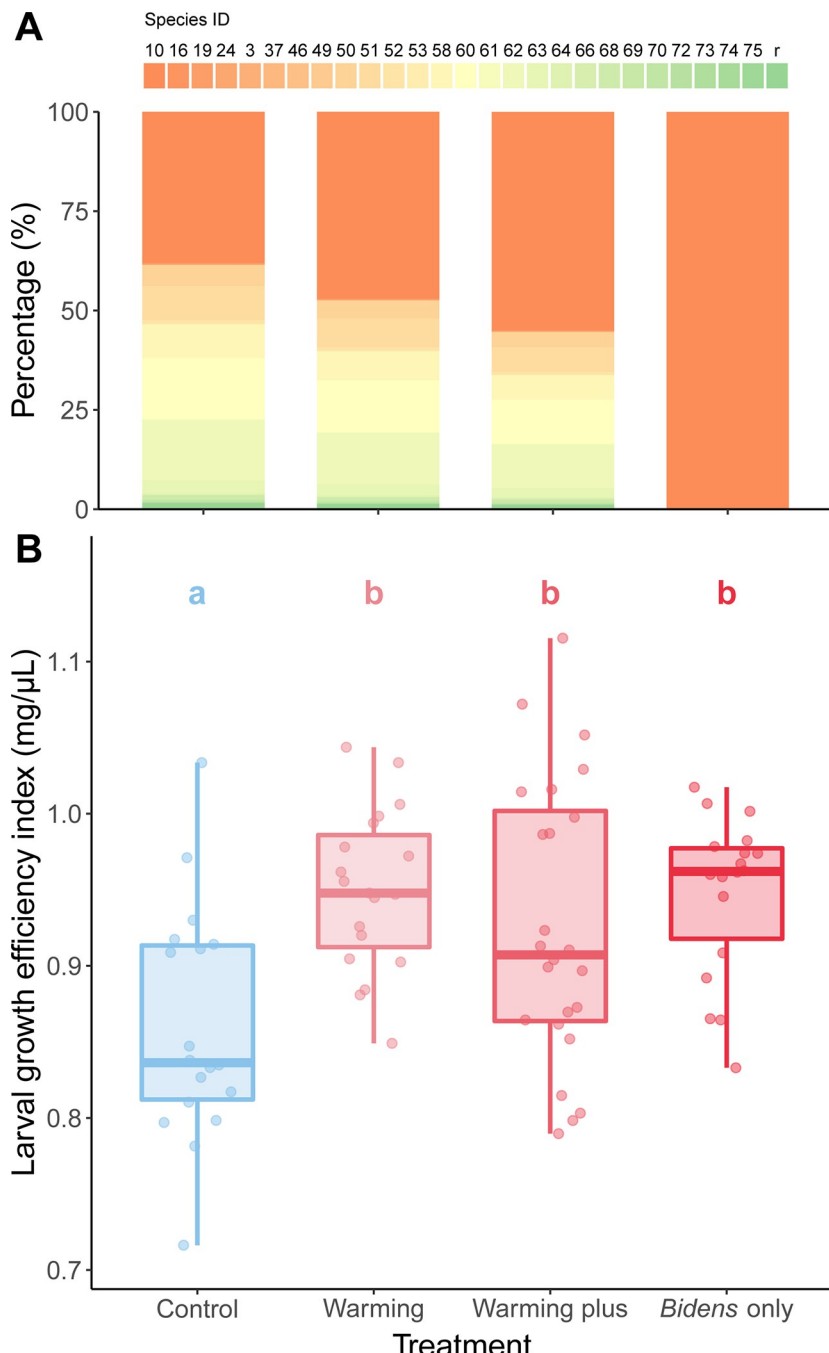

**Fig 4. The efficiency of bee larval growth when fed with semi-artificial diet.** (A) Bee pollen composition in four treatments: control, warming, warming plus and B. pilosa only, which were fed to bees in different proportions. Different colors represent different species of pollen. (B) The box plot shows bee larval efficiency index, with the medians (center lines), interquartile ranges (boxes), and the largest and smallest values (whiskers) within 1.5 times the interquartile range. Each point denotes a single larva. The letters denote significant statistical difference among the treatments.

**Table 2. Post-hoc comparison of larval efficiency index in different treatments.** The efficiency index in the warming, warming plus and *B. pilosa* only groups were all significantly better than the control group.

| Contrast | Estimate | z | p |
|---|---|---|---|
| Control—Warming | -0.10 | -3.90 | **<0.001** |
| Control—Warming plus | -0.07 | -2.98 | **0.016** |
| Control—*B. pilosa* only | -0.10 | -3.74 | **0.001** |
| Warming—Warming plus | 0.03 | 1.15 | 0.657 |
| Warming—*B. pilosa* only | 0.003 | 0.11 | 1.000 |
| Warming plus—*B. pilosa* only | -0.03 | -1.02 | 0.737 |

*p* values <0.05 are highlighted in bold.

## Discussion

Our results showed that warming does change bee foraging behavior within the day, with more foragers departing from beehives and carrying back more pollen in the early morning [21, 48].

Nonetheless, pollen collection rate per individual of bees remained unaffected by warming. We further showed that warming led to a shift of pollen release by an hour earlier in *Bidens pilosa*, the most abundant and widely distributed plant species in our study site. This earlier shift of anthesis in *B. pilosa* may lead to changes in available pollen sources, particularly in the early morning, potentially affecting the composition of pollen that is fed to the larvae.

By manipulating pollen compositions under projected future climate warming scenario (e.g., increased *B. pilosa* pollen percentage), our larval development experiment showed that such composition change increased bee larval growth efficiency, highlighting the potential role of climate change in bee pollen diversity and larval performance.

Environmental factors that are known to affect bee foraging—such as temperature and solar radiation—vary throughout the day [49]. In accordance with previous study [24], we observed that foraging activity and pollen collection peaked in the early morning, with the number of foragers and the weight of pollen loads increasing with temperature. However, pollen collection rate did not correlate with temperature. This is likely because our experimental warming was applied only to the hives and not to the entire surrounding environment where foraging occurs. Consequently, the initial metabolic benefits gained from a warmer hive are quickly mitigated when bees are exposed to the ambient temperatures outside the hive. A more comprehensive understanding of the relative importance of within-hive and ambient temperature to bee foraging behavior could be developed by considering multiple environmental factors in mediating bee foraging behaviors. Warming enhanced foraging at 6:00, especially at temperatures below 21°C (Fig 2; S5 Fig in S1 File), but not above 27°C, likely because our study's average temperature ranged between 21–27°C, and so the effect was not captured by the range of temperatures in this study. We found that the pollen collection rate decreased with time, irrespective of the temperature treatments. We speculate that bees could increase the number of water and nectar foragers during hotter afternoons, resulting in decrease in pollen collection rate [48, 50], although we were unable to differentiate this behavior based on our video analysis. Additionally, since early-foraging bees may have already collected substantial amounts of the available pollen, later foragers may find reduced overall pollen quantities, thereby decreasing their pollen collection rate.

Our analysis of pollen composition showed that *B. pilosa* was the most frequently present plant species (S3A Fig in S1 File). Although honey bees are generalist pollinators [51], they can be particularly attracted to plant species that bloom in high abundance, such as *Koelreuteria*

*elegans* and *Bauhinia x blakeana*. The variations in pollen composition reflect the bees' ability to adapt their foraging strategies based on locally available floral resources [52]. Depending on the types of floral resources (pollen or nectar) and the floral landscape (e.g., the most abundant or closest resource to the hive), bees make decision of visiting specific patches of flowers [26]. This temperature-mediated floret anthesis of *B. pilosa* can lead to changes in the pollination of other plants, especially those that normally bloom earlier than *B. pilosa*, such as *Acacia confusa*, *Melaleuca leucadendra/quinquenervia* and *Polyspora axillaris* (S6 Fig in S1 File). Nevertheless, we did not test the effect of warming on the time of anthesis in other plant species but specifically focused on *B. pilosa*. This is because *B. pilosa* was the species with the highest pollen abundance in our study area, and because *B. pilosa* blooms year-round and is mostly available to bees. We acknowledge, however, that our findings might not fully represent the dynamics of other native plant species, which could also be significantly affected by warming [53, 54]. Given that warming likely influences a wide array of plant species, future studies will need to comprehensively assess how changes in temperature affect the pollen loads across entire plant communities.

Finally, our larval development results showed that an increase in the pollen percentage of *B. pilosa*, as projected from a future warming scenario, promoted bee larval development. This outcome contradicts our hypothesis that a less diverse or less nutritious diet would negatively influence larval development. The nutrients of pollen, including essential proteins, lipids and vitamins [55], plays a crucial role in supporting bee growth and development. Among these nutrients, protein level is a key determinant of the lifespan of bees [56] and brood rearing success [57]. Therefore, while a less diverse pollen composition may limit the larval growth and development of bees due to the lack of necessary elements [29], studies suggest that the availability of particular pollen species rich in protein content may compensate for reduced diversity of the flora for larval growth and development [31]. Another study suggests that the nutritional quality combined with diversity of pollen nutrition can maintain the health of bees [30]. Unfortunately, logistical constraints prevented us from simultaneously warming both hives and plants, an approach that would be more realistic and warrants future investigation.

The complex structure of pollen walls, including species-specific variations in porosity and thickness, affects its digestibility by honey bees [55]. For example, the pollen of *B. pilosa* from the family Asteraceae has a particularly tough wall that limits its digestibility [58], which could lead to a lower nutritional value [59], and, could potentially result in lighter bee larvae [60]. However, our findings contrast with those of Tasei and Aupinel (2008), as we observed increased larval mass with higher proportions of *B. pilosa* pollen in the diet. This discrepancy could be due to the role of endosymbiotic bacteria, transmitted by nurse bees [61], which aid in breaking down the pollen wall [62]. However, the composition and abundance of endosymbiotic bacteria can change across several factors, such as landscapes [63] and seasons [63, 64]. Since three months passed between our field and larvae feeding experiments, the endosymbionts responsible for digesting pollen during the larval development experiment might have changed to target different plant species that were more common compared to those during the field experiment.

The consistent availability of *B. pilosa* throughout different seasons might support a stable gut microbiome in bees by providing a reliable source of specific nutrients that certain endosymbiotic bacteria utilize [65]. Further research should align pollen collection with larval feeding to clarify the impact of seasonal endosymbiont and pollen changes on bee development. Although larvae attained a higher body mass when fed a diet with an increased proportion of *B. pilosa* pollen, a pollen-only diet may impact larval resilience by potentially limiting the diversity of nutrients and secondary metabolites that larvae would otherwise obtain from a

mixed diet. This limited nutrient diversity might weaken their immune responses, making them more susceptible to stressors like viruses, pesticides, or parasites [66], suggesting that adult bee health and other developmental factors, such as larval immune responses and gut microbiota diversity, need to be evaluated for a more comprehensive understanding of the consequences of larval development under varying dietary compositions.

In this study, we acknowledge the limitation of using only two bee hives, one as a warming treatment whereas the other as a control. This setup raises the possibility that hive-specific factors, apart from temperature, could influence the observed differences in foraging behavior. To mitigate this potential hive-specific bias, we employed an experimental design where the warming treatment was alternated between the two hives. Additionally, this alternation also minimized the potential for acclimatization effects rather than merely short-term physiological changes induced by warming. This allows for a more focused investigation of temperature effects on behaviors, despite the fact that the residual effects of warming might persist in hives even after switching treatments. To minimize the potential influence of pseudoreplication, our detailed and repeated observations over the study period may account for a more nuanced understanding of hive responses to warming. We have also considered beehive as a random effect to account for the inherent variability within each hive. Together, this approach aimed to ensure that any observed effects could be more confidently attributed to temperature changes rather than to characteristics inherent to a particular hive. However, future studies should consider including a larger number of hives to test the robustness of our findings to enhance the robustness of the conclusions drawn from similar experimental setups.

Over the past decade, the effects of climate change, particularly global warming, have increasingly raised concerns about the stability of ecosystems and biodiversity [1]. This has heightened the importance of understanding how warming impacts plant-pollinator interactions, as shifts in these interactions can lead to cascading effects on ecosystem services [67]. While most studies have focused on seasonal phenological mismatches induced by temperature warming, our study suggests that temperature changes at a short-term scale can be just as important in influencing foraging behaviors in bees and the time of plant anthesis. These short-term shifts may have implications for potential changes in pollen composition, which in turn, could shape the fitness consequences of larval development.

## Supporting information

**S1 File.**
(DOCX)

## Acknowledgments

We thank Jian-Hui Li, Qiu-Ling Zhao, Jie Ding, Yun-Ru Chen, Po-Hsiung Lin, Chien-Chih Yang, Ying-Lien Chen, Li-Jie Wang, Yun-Chen Hsieh, Zi-Ying Liu, and Wen-Ting Chen for logistic supports. We thank the other members of the Chuan-Kai Ho Group. We also thank two anonymous reviewers for their insightful comments. Last but not least, we thank the bees.

## Author Contributions

**Conceptualization:** Megan M. Y. Chang, Syuan-Jyun Sun, Chuan-Kai Ho.

**Data curation:** Megan M. Y. Chang, Pei-Shou Hsu, Syuan-Jyun Sun.

**Formal analysis:** Megan M. Y. Chang, Pei-Shou Hsu, Syuan-Jyun Sun.

**Funding acquisition:** Chuan-Kai Ho.

**Investigation:** Megan M. Y. Chang, Pei-Shou Hsu, Syuan-Jyun Sun, Chuan-Kai Ho.

**Methodology:** Megan M. Y. Chang, Pei-Shou Hsu, En-Cheng Yang, Syuan-Jyun Sun, Chuan-Kai Ho.

**Project administration:** Syuan-Jyun Sun.

**Resources:** Pei-Shou Hsu, En-Cheng Yang, Syuan-Jyun Sun, Chuan-Kai Ho.

**Software:** Megan M. Y. Chang, Syuan-Jyun Sun.

**Supervision:** Syuan-Jyun Sun, Chuan-Kai Ho.

**Validation:** Megan M. Y. Chang, Pei-Shou Hsu, Syuan-Jyun Sun.

**Visualization:** Megan M. Y. Chang, Syuan-Jyun Sun.

**Writing – original draft:** Megan M. Y. Chang, Pei-Shou Hsu, Syuan-Jyun Sun, Chuan-Kai Ho.

**Writing – review & editing:** Megan M. Y. Chang, Pei-Shou Hsu, En-Cheng Yang, Syuan-Jyun Sun, Chuan-Kai Ho.

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
