## [Decision Letter · Decision Letter 0]

18 Oct 2024

PONE-D-24-21139Warming induces short-term phenological shifts in pollinator-plant interactions that enhance larval development in honey beePLOS ONE

Dear Dr. Sun,

Thank you for submitting your manuscript to PLOS ONE. After careful consideration, we feel that it has merit but does not fully meet PLOS ONE’s publication criteria as it currently stands. Therefore, we invite you to submit a revised version of the manuscript that addresses the points raised during the review process.

We look forward to receiving your revised manuscript.

Kind regards,

Rachid Bouharroud

Academic Editor

PLOS ONE

Journal Requirements:

plos.org/plosone/s/file%3fid=wjVg/PLOSOne_formatting_sample_main_body.pdf%20and">When submitting your revision, we need you to address these additional requirements.

4. Thank you for stating the following financial disclosure: “National Science and Technology Council 108-2621-B-002-003-MY3 (CKH)

National Science and Technology Council 111-2621-B-002-003-MY3 (CKH)”

. Please state what role the funders took in the study. If the funders had no role, please state: "The funders had no role in study design, data collection and analysis, decision to publish, or preparation of the manuscript." If this statement is not correct you must amend it as needed. Please include this amended Role of Funder statement in your cover letter; we will change the online submission form on your behalf.

Additional Editor Comments:

Dear Authors

Please check review comments for both reviewers with more attention to Reviewer 1 as his comments will deeply improve your manuscript.

Good luck

Reviewers' comments:

Reviewer's Responses to Questions

**Comments to the Author**

1. Is the manuscript technically sound, and do the data support the conclusions?

Reviewer #1: Partly

Reviewer #2: Partly

2. Has the statistical analysis been performed appropriately and rigorously? 

Reviewer #1: No

Reviewer #2: Yes

3. Have the authors made all data underlying the findings in their manuscript fully available?

Reviewer #1: Yes

Reviewer #2: Yes

4. Is the manuscript presented in an intelligible fashion and written in standard English?

Reviewer #1: Yes

Reviewer #2: Yes

5. Review Comments to the Author

Reviewer #1: This paper represents an interesting and novel contribution to our understanding of how climate change-induced warming may influence honeybee behavior and fecundity, as well as flowering timing. The study provides evidence that in the early morning, warming will cause honey bees to forage at higher rates and at higher efficiencies relative to controls, but that warming has no effect on the composition of pollen collected by honeybees. In addition, the study demonstrates that warming advances peak anthesis timing. Finally, it shows that a higher ratio of pollen from a focal species improves larval efficiency. To my knowledge, this is the first paper to address the impact of warming on the time of day of honeybee activity and their primary forage species within the system.

However, I feel that there are several issues with the current version of the manuscript that need to be addressed prior to publication. Principally, the design of the study and language throughout the paper seem to imply that increased temperature will coincide with an increase in prevalence of the pollen of Bidens pilosa; however, to my knowledge, this connection was not directly studied. Second, I am concerned that there was only a single beehive used for each treatment. Though I recognize that this was in part ameliorated by the alternation of warming treatments, it remains an issue that is insufficiently addressed in the body of the manuscript. Third, the statistical approaches taken in some areas of the analysis could be improved. Finally, I have several line edits that I have separated by section that may help improve the flow and readability of the manuscript.

I thank the authors for their consideration of these comments. I hope to help assist with this manuscript again in the future.

Please review my comments in the attached document.

Reviewer #2: The manuscript has significant value in bee management. However, vigorous revision is needed. A table containing a list of bee-foraged plants (with relative frequencies) will enrich the manuscript. My specific comments are given below:

Line 50: Pollinators and plants must match phenologically----- mean? Rewrite clearly. Diverse insect species pollinate most flowering plants. However, all are not equally effective. The pollination efficiency of a pollinator depends on its foraging strategy and floral characteristics (Layek et al. 2022 Journal of Asia-Pacific Entomology 25: 101882. Doi: 10.1016/j.aspen.2022.101882).

Line 51-52: Bees (Hymenoptera: Anthophila) are one of the most ecologically and economically important pollinators worldwide (Hristov et al., 2020)----- add more references.

Line 158-159: results showed that warming does change bee foraging behavior within the day, with more foragers departing from beehives and carrying back more pollen in the early morning. ----- Much literature is available regarding this issue (e.g., Forcone et al. 2011 Grana 50: 30-39. Doi: 10.1080/00173134.2011.552191; Tan et al. 2012 Apidologie 43: 618-623. doi 10.1007/s13592-012-0136-y; Layek et al. 2020 Palynology 44: 114-126. Doi: 10.1080/01916122.2018.1533898). Consider these.

Line 172-173: pollen weight increasing with temperature----- Weight of individual corbicular pollen load? Or total pollen collection per unit of time? It needs to be clearly described throughout the manuscript. The amount of pollen collection greatly depends on the hive's surrounding pollen availability and the colony’s demand.

Line 160: foraging efficiency per individual of bees remained unaffected by warming. ----- Did they measure the foraging efficiency of an individual worker? If yes, it needs to be detailed in the Material & method and Result section.

Line 293: eight 5-minute observations each day of the experiment------ eight observations taken throughout the daytime at regular intervals?

Line 304-306: For each observation, we also determined the foraging efficiency per individual by dividing pollen weight by the number of foragers recorded from each 5-minute footage.---- It is just the number of returning pollen foragers. All foragers do not collect pollen at a given time. So, it is not actual foraging efficiency. Therefore, avoid the term ‘foraging efficiency’.

Line 423-469: statistical analysis: Rewrite more compactly.

6. PLOS authors have the option to publish the peer review history of their article (what does this mean?). If published, this will include your full peer review and any attached files.

Reviewer #1: No

Reviewer #2: No

---

## [Author Response · Author response to Decision Letter 0]

4 Nov 2024

5. Review Comments to the Author

Reviewer #1: This paper represents an interesting and novel contribution to our understanding of how climate change-induced warming may influence honeybee behavior and fecundity, as well as flowering timing. The study provides evidence that in the early morning, warming will cause honey bees to forage at higher rates and at higher efficiencies relative to controls, but that warming has no effect on the composition of pollen collected by honeybees. In addition, the study demonstrates that warming advances peak anthesis timing. Finally, it shows that a higher ratio of pollen from a focal species improves larval efficiency. To my knowledge, this is the first paper to address the impact of warming on the time of day of honeybee activity and their primary forage species within the system.

However, I feel that there are several issues with the current version of the manuscript that need to be addressed prior to publication. Principally, the design of the study and language throughout the paper seem to imply that increased temperature will coincide with an increase in prevalence of the pollen of Bidens pilosa; however, to my knowledge, this connection was not directly studied. Second, I am concerned that there was only a single beehive used for each treatment. Though I recognize that this was in part ameliorated by the alternation of warming treatments, it remains an issue that is insufficiently addressed in the body of the manuscript. Third, the statistical approaches taken in some areas of the analysis could be improved. Finally, I have several line edits that I have separated by section that may help improve the flow and readability of the manuscript.

I thank the authors for their consideration of these comments. I hope to help assist with this manuscript again in the future.

Please review my comments in the attached document.

We are very grateful for reviewer 1 for the very constructive comments. We have accepted most of them with suggested changes. We have also elaborated in details our study design and discussed potential issue of pseudoreplication. We have now responded each specific comment below. 

Title and abstract: 

Reviewer comment 1:

The title and abstract are mostly well written, but I feel that they overstate the results of this paper. Though it is used throughout the title and abstract, I believe that the term ‘phenological shifts’ is only tenuously connected to the findings of this study. The term ‘timing’ may be more appropriate in some cases. In addition, given that composition of pollen collected by bees did not vary significantly between treatments and that both bee and flower timing were advanced by warming, I am unsure as to why the authors state that climate change may affect larval development by changing pollen provisioned to them. 

Author response 1: Thank you for your insightful comment. We agree that ‘phenological shifts’ traditionally refers to longer-term, seasonal changes. However, our study addresses shifts in the short-term synchrony of biological timing events (e.g., timing of bee foraging and flower anthesis), and thus, we used ‘phenological shifts’ to capture the interconnected timing changes in bees and flowers under warming conditions. We have further clarified our terminology in the revised manuscript, particularly where ‘timing’ is more appropriate. 

We also appreciate your concern regarding the interpretation of pollen composition and larval development. Since we found no significant difference in pollen composition between control and warming treatments, we understand that discussing pollen-driven impacts on larval development may have been unclear. In our study, we found the advanced effect of warming on the timing of bee foraging (Experiment 1) and flowering (Experiment 2), with the expectation that earlier flowering induced by warming would lead to higher availability of B. pilosa pollen for bees during earlier foraging hours. Subsequently, we manipulated pollen compositions that reflect a projected future scenario (Experiment 3) to assess their impacts on larval development. Unfortunately, logistical constraints prevented simultaneous warming of both hives and plants, which we acknowledge as a limitation of our study. 

Reviewer comment 2:

L21: ‘altering their phenologies’- please clarify if both or one taxa’s phenologies are being altered

Author response 2: By this we meant both the phenologies of plants and pollinators can be altered. We have now made this clear by adding ‘of both parties’ after ‘altering their phenologies’.

Reviewer comment 3:

L22: ‘short-term’- here, it may be helpful to clarify what length of time is being referred to. E.g. ‘within-day’ 

Author response 3: Thank you for your comment. We have rephrased this sentence by adding long-term, seasonal phenological changes, as a contrast to short-term phenological changes. Yet we prefer to use short-term here, but not specify ‘within-day’, since we would like the abstract to begin as general as possible. In addition, we have specified our scale as ‘within-day’ in the following sentences.

Reviewer comment 4:

L23: ‘this issue’- rephrase to be more specific

Author response 4: We have now combined this sentence and the next to make our case specific. 

Reviewer comment 5:

L26: ‘pollen composition’- this term is used very often throughout the manuscript, and should be clarified in the abstract and main text. Malagnini et al. (2022) do a great job of doing this (https://doi.org/10.3389/fsufs.2022.865368)

Author response 5: We agree that it needs to be clarified upfront. We have made it clearer by adding ‘, in terms of the relative abundance of pollen from different plant species’.

Reviewer comment 6:

L28: ‘studied’- the authors did not directly study these plants, so it may be better to clarify that these were the most commonly represented species in the pollen collected by bees. Also, I suggest adding a smoother transition between these sentences, like “Further” 

Author response 6: We have now rephrased this sentence.

Reviewer comment 7:

L29: ‘in the diet’- add ‘pollen’ before this

Author response 7: We have now added ‘pollen’.

Reviewer comment 8:

L30-31: I am unclear on how this study provides evidence for a disruption in interactions, as the timing of both species was advanced

Author response 8: We agree with you that in our case, disruption is not appropriate. Yet since we have only tested specifically the interactions between bees and B. pilosa, what we found here might not be applicable to bee’s interactions with other plant species. Thus, we have rephrased this sentence accordingly. 

Reviewer comment 9:

L30: If this sentence structure is maintained, I suggest changing ‘highlights’ to ‘demonstrates’

Author response 9: We have changed ‘highlights’ to ‘demonstrates’, which is more appropriate.

Reviewer comment 10:

L31: Suggest changing ‘phenology’ to ‘timing’

Author response 10: We thank the reviewer for this comment and prefer to retain the term ‘phenology’ as phenology specifically refers to the timing of biological events, such as flowering and pollinating, in response to environmental changes.

Reviewer comment 11:

L32: Omit ‘the fitness of’ 

Author response 11: We have removed ‘the fitness of’.

Introduction: 

Reviewer comment 12:

The introduction is mostly well-written and clear. However, the first and second paragraphs of the introduction could be distilled and merged as there are several redundancies among them. Alternatively, they could be made more disparate by emphasizing certain concepts in each paragraph. Further, the fourth paragraph emphasizes phenological shifts, but does not discuss how climate change may influence within-day foraging and flowering timing. The authors may benefit from looking for literature on diel rhythms, 24-hour timing, and circadian clocks of pollinators and flowers to provide more thorough background on the drivers of insect activity and flowering timing. For example, please see the following papers: 

Bloch et al. 2017- https://doi.org/10.1098/rstb.2016.0256 Steen et al. 2016- https://doi.org/10.1111/2041-210X.12654 Auffray et al. 2017- https://doi.org/10.1093/jisesa/iex018

Other several important parts of the study are insufficiently supported by the introduction, including the drivers of pollen load size and plant species composition, as well as the influence of diet on larval development efficiency. These concepts need to be addressed in the introduction. Please find my line edits below. 

Author response 12: Thank you for your valuable feedback! We have carefully followed your suggestions here and in your additional comments to streamline the structure by merging paragraphs and minimizing redundancies. We have expanded the third paragraph in the introduction to illustrate the links between phenological shifts and the potential impact of climate change on plant-pollinator interactions. Additionally, we have strengthened the introduction by providing more background on the factors influencing the number of foragers, pollen load weight, and pollen composition, as well as their implications for larval development. This added context supports our focus on these critical aspects of the study. 

Reviewer comment 13:

L39: Double space before ‘(Traill’

Author response 13: We have made it a single space.

Reviewer comment 14:

L42-43: This is phrased to imply that general disruptions to plants and pollinators cause phenological mismatches- suggest rephrasing to ‘...interactions by causing phenological mismatches.’

Author response 14: We have rephrased it as you suggested.

Reviewer comment 15:

L46-47: I suggest rephrasing to ‘A reduction in effective pollination could reduce seed set in plants that rely on pollinator outcrossing for reproduction, thereby reducing the total quantity of pollen and nectar rewards available to pollinators in future seasons’, then cite this concept. L50: ‘effectively increase’ to ‘facilitate’

Author response 15: We appreciate your suggested changes and make the subsequent changes, which flows better now; in previous L50, this sentence does not exist anymore and so this comment is not relevant.

Reviewer comment 16:

L50-51 and L55-62: These sentences are more related to the sentences on L42-45 than those on L51-52. If L50-51 and L55-62 are merged into the first introduction paragraph, flow would improve. Alternatively, the authors could introduce the idea that climate change causes phenological mismatches and detriments pollination in paragraph 1, then emphasize its effect on honeybees in paragraph 2.

Author response 16: Thank you for your suggestion. We have considered this modification by integrating L55-62 into paragraph 1. We have also removed previous L50-51 since we have already presented similar concept in paragraph 1 to avoid repetition.

Reviewer comment 17:

L58: I suggest adding ‘or be reduced’ after ‘coincide’

Author response 17: We have made the suggested change.

Reviewer comment 18:

L64: I suggest adding ‘most’ after ‘To date,’

Author response 18: We have added ‘most’ after ‘To date’.

Reviewer comment 19:

L65: Please clarify the meaning of ‘spring activities’

Author response 19: We have replaced the previous reference on bumble bees with an updated ones of honey bees, and have made this term clear.

 Reviewer comment 20:

L67: ‘fluctuations’- please describe what fluctuations these refer to. Also, I’m not sure that it is entirely true that these responses remain unclear- the next sentence appears to address some of these responses. It may improve clarity to rephrase to say that these responses are understudied.

Author response 20: We agree with you that ‘fluctuations’ is unclear. We have now made it clear as ‘within-day phenological match-mismatch’. We have also replaced unclear with understudied.

Reviewer comment 21:

L70-70: I suggest changing the part of this sentence after ‘et al., 2020)’ to ‘causing bees to seek water for use in evaporative cooling’

Author response 21: We have made the suggested change to this sentence.

Reviewer comment 22:

L71: ‘The foraging activity’ of what species?

Author response 22: We have added A. mellifera.

Reviewer comment 23:

L72: Add ‘within-day’ before ‘variations

Author response 23: We have added ‘within-day’.

Reviewer comment 24:

L74-75: This same concept was stated on L69-70

Author response 24: We have removed the sentence (previous L74-75).

Reviewer comment 25:

L75-76: Repetitive with previous sentences

Author response 25: We have removed this sentence.

Reviewer comment 26:

L76-80: This idea would be more useful if it were tied it into the hypothesis on L104-106

Author response 26: We have now referred this work done by Reddy et al. in our hypothesis.

Reviewer comment 27:

L78: Please cite (Reddy et al. 2015) here as well

Author response 27: We have now cited it.

Reviewer comment 28:

L80-82: This feels contradictory to the phrase on L66-67

Author response 28: We have rephrased L80-82.

Reviewer comment 29:

L82-83: I suggest developing this idea more to provide more support for the anthesis onset component of the paper

Author response 29: We noticed that our use of anthesis onset is inappropriate for our focal species B. pilosa, which can be misleading; thus, we have replaced this term with flowering.

Reviewer comment 30:

L86: This may be a good place to describe the meaning of pollen composition (see my comment about L26)

Author response 30: We have defined our use of pollen composition as the variety of bee pollen collected by bees from different plant species.

Reviewer comment 31:

L87: This is setting the reader up for a paper on phenology, not on daily fluctuations in resource availability

Author response 31: Our purpose of setting up this paragraph, focusing on plant phenology, is to act as a contrast of previous paragraphs, which focus on temperature change effect on pollinator activities. Hence, we do not intend to narrow this idea down here to mean daily fluctuations but plant phenology in general.

Reviewer comment 32:

L92: This raises the question of whether honeybees eat pollen or if they only use it to provision their larvae- this would be an interesting tidbit to add

Author response 32: Thank you for raising this point. Although we mentioned the role of pollen on bee physiology and larval development here, only larvae feed on the pollen (not adults). By bee physiology we meant that earlier diet of pollen can determine bee physiology as they become adults. We have now clarified this.

Reviewer comment 33:

L93: I suggest changing ‘study experimentally manipulating’ to ‘a study that experimentally manipulated’

Author response 33: Thank you – we have made the suggested change.

Reviewer comment 34:

L101-103: These questions may better prepare the reader to understand this paper if they are edited to align directly with the individual components of the experiment. In addition, I suggest rephrasing these to questions- e.g., ‘(1) does warming affect bee foraging behaviors

Author response 34: Thank you for the comments. We have rephrased these into three questions. Also, we have referred these three questions alongside the respective experiment 1-3.

Reviewer comment 35:

L104: Meaning of ‘through’ unclear

Author response 35: We have replaced ‘through’ with ‘by altering’.

Reviewer comment 36:

L104-106: Add ‘would’ after ‘warming’ and describe what aspect of foraging behavior would increase/decrease

Author response 36: We have added ‘would’ after ‘warming’. We have also made clear in the predictions that our use of foraging behavior here refers to the number of foragers and the weight of pollen loads carried by bees. 

Reviewer comment 37:

L109: ‘of’ to ‘in’ and ‘predict’ to ‘expected to find’ 

Author response 37: We have made these two changes.

Results: 

Reviewer comment 38:

The results are well organized. However, I am confused by the interpretation of some model results, especially those on L117-127. I do not

---

## [Editor Report · Decision Letter 1]

18 Nov 2024

Warming induces short-term phenological shifts in pollinator-plant interactions that enhance larval development in honey bee

PONE-D-24-21139R1

Dear Dr. Sun,

We’re pleased to inform you that your manuscript has been judged scientifically suitable for publication and will be formally accepted for publication once it meets all outstanding technical requirements.

Kind regards,

Rachid Bouharroud

Academic Editor

PLOS ONE
---

## [Editor Report · Acceptance letter]

20 Nov 2024

PONE-D-24-21139R1 

PLOS ONE

Dear Dr. Sun, 

I'm pleased to inform you that your manuscript has been deemed suitable for publication in PLOS ONE. Congratulations! Your manuscript is now being handed over to our production team.

Kind regards, 

on behalf of

Dr. Rachid Bouharroud 

Academic Editor

PLOS ONE